# Cold Shock Disrupts Massed Training-Elicited Memory in Drosophila

**DOI:** 10.3390/ijms23126407

**Published:** 2022-06-08

**Authors:** Anna Bourouliti, Efthimios M. C. Skoulakis

**Affiliations:** 1Institute for Fundamental Biomedical Research, Biomedical Sciences Research Center “Alexander Fleming”, 16672 Vari, Greece; bourouliti@fleming.gr; 2Department of Molecular Biology and Genetics, Democritus University of Thrace, 68100 Alexandroupolis, Greece

**Keywords:** memory, anesthesia resistant memory, olfactory conditioning, massed conditioning, Drosophila

## Abstract

Memory consolidation is a time-dependent process occurring over hours, days, or longer in different species and requires protein synthesis. An apparent exception is a memory type in *Drosophila* elicited by a single olfactory conditioning episode, which ostensibly consolidates quickly, rendering it resistant to disruption by cold anesthesia a few hours post-training. This anesthesia-resistant memory (ARM), is independent of protein synthesis. Protein synthesis independent memory can also be elicited in *Drosophila* by multiple massed cycles of olfactory conditioning, and this led to the prevailing notion that both of these operationally distinct training regimes yield ARM. Significantly, we show that, unlike *bona fide* ARM, massed conditioning-elicited memory remains sensitive to the amnestic treatment two hours post-training and hence it is not ARM. Therefore, there are two protein synthesis-independent memory types in *Drosophila*.

## 1. Introduction

Unconsolidated memories are labile and disrupted by amnestic agents in all animals tested [1,2]. In Drosophila, a brief cold shock immediately following negatively reinforced olfactory conditioning results in complete memory loss of the association. However, if delivered a couple of hours post-training it is incompletely disruptive, with the residual memory termed anesthesia resistant memory (ARM), as it persists in the apparently anesthetic, immobilizing cold shock [3] and is independent of protein synthesis [2,4]. ARM appears to consolidate relatively rapidly as it is partially labile minutes after conditioning [5] and stable by 2 h post-training [3,6]. A protein synthesis independent memory also emerges after multiple consecutive rounds (massed conditioning-MC) of negatively reinforced olfactory conditioning [3]. Although cold shock treatment after a single round of conditioning is typically used to probe 3-h memory and the MC protocol is utilized for 24-h memory assessment, the presumed protein synthesis independence has led to these two memory types being called ARM.

Despite evidence suggesting that both of these operationally and temporally distinct memory types engage common, or similar mechanisms [4,7,8], it is unclear whether they reflect the same cognitive outcome assayed at different time points or are in fact distinct memory types. While both cold shock and five or ten-round MC protocols are widely used to address questions regarding ARM [3,4,7], data acquired by one training method are not typically cross-checked with the other. To elucidate whether memory with the same properties is formed with both methods, we hypothesized that if the ARM is equivalent or the same as MC yielded 24-h memory, a cold shock delivered 2 h post-training should have a similar effect on both. However, this notion would not be supported if this amnestic treatment impairs 24-h MC memory. To this end, we conditioned wild-type Drosophila to associate an aversive odor with electric foot shock by training with five MC rounds and subjected them to a single cold shock prior to testing, or 2 h after training. We have been using a five-round MC [4] because in our hands it affords higher resolution than more intensive protocols that may yield “ceiling” effects. We report that 2 h post-training, five-round MC yields a cold-shock sensitive memory along with ARM.

## 2. Results

Conditioning was performed using the classical aversive conditioning paradigm which pairs an aversive odor (conditioned stimulus-CS+) with electric foot shocks (the unconditioned stimulus-US), while a second equally aversive odor explicitly unpaired with the foot shock (CS−) serves as a control [9]. Memory immediately after one conditioning round contains an ARM component [5,10]. However, the defining brief cold shock typically used to reveal the non-labile ARM memory component 3 h after a single round of conditioning [3,11], has not been applied after five rounds of MC to our knowledge. Cold shock immediately after one round of training disrupts memory completely [11]. However, this immediate effect cannot be addressed in our experiments as the time it takes to deliver five training rounds unavoidably leads to testing twenty minutes after the first foot shock/odor association. Therefore, in all our experiments cold shock is delivered at the indicated times after the last round of training.

As shown in Figure 1A, 8-min memory after five rounds of MC is also largely labile and disrupted by a brief cold shock. Unexpectedly, a 2-min cold shock 2 h after five rounds of MC resulted in a significant reduction of 3-h memory (Figure 1B). This effect is not specific to the *w^1118^* strain, as an identical memory decrease was uncovered in Canton S flies (Figure 1B). Therefore, memory elicited by five rounds of MC does not yield solely ARM, but rather both cold-shock-sensitive and cold-shock-insensitive memory components. It follows then, that consolidated memory after MC is not the same as that after a single round of training. Unsurprisingly, given the kinetics of the protein synthesis sensitive long-term memory (LTM) [3], five rounds of spaced conditioning, a paradigm where the training rounds are spaced apart by fifteen minutes, which leads to LTM formation [2,3,6], also yielded a consolidated and a labile memory component 3 h post-training (Figure 1C). This suggests that both massed and spaced conditioning elicited memories are composed of labile components even at 3 h post-training. The residual memory persisting amnestic treatment after spaced training is likely ARM as suggested previously [3].

Are consolidated memories 2 h post-conditioning after a single or five MC rounds, equivalent, or proportional to the training intensity? To directly compare the memory levels, we trained flies with either one round or five MC rounds and administered cold shock 2 h later. The 3-h memory of untreated five MC trained animals was significantly different from that of one-round-trained flies (Figure 1D), indicating that the intensive MC training leads to a more robust three-hour memory. However, cold-shock-resilient memories were not significantly different, suggesting that the consolidated ARM component is not affected by training intensity. The above observations were also apparent when the relative non-consolidated post-cold-shock memory was calculated, which demonstrated that although there is no statistically significant difference between the two paradigms, the mean absolute value of memory decline is higher than that yielded by the five-round MC (Figure 1E). This likely reflects the elevated labile memory yielded by MC.

We hypothesized that the cold-shock-sensitive component after MC may reflect memory consolidating with slow kinetics. Hence, a cold shock was administered 1 h before 24-h memory assessment after MC to investigate memory stability at this point. This pre-testing cold shock did not affect memory, as the performance was not different from similarly trained flies not subjected to the amnestic treatment (Figure 2A). This result demonstrates that MC-elicited memory was consolidated at 23 h post-training and verifies that cold-shock treatment does not generally affect recall. In contrast, a cold shock delivered 2 h after a five-round MC resulted in significantly reduced 24-h memory compared to that of non-cold shocked flies or animals subjected to the amnestic treatment 1 h prior to testing (Figure 2B). Therefore, memory elicited by a five-round MC includes a significant labile component at 2 h post-training, which when blocked is reflected in compromised 24-h olfactory associative memory. Collectively, these results strongly suggest that a five-round MC yields a memory type that consolidates slowly, being labile at 3 h, whereas ARM is not [3].

## 3. Discussion

Massed conditioning (MC) in a negatively reinforced olfactory conditioning task yields a translation-independent 24-h memory. Since 1995, when the protocol was first reported, it has been assumed that massed conditioning-yielded memory is equivalent to 3-h memory elicited by a single round of conditioning and revealed as resilient to a cold shock 2 h post-training. The 3-h ARM and MC-elicited 24-h memory are both independent of protein synthesis [2,3] and are reported to engage common molecular components [3,4,12]. Clearly, however, unlike ARM, five-round MC-elicited memory is not resistant to amnestic treatment 2 h post-training (Figure 1E). Moreover, the amnestic treatment at 2 h post-training nearly eliminates 24-h memory of the event, further supporting the notion that, unlike ARM, this MC elicited memory is not consolidated at that time.

Therefore, MC elicits a distinct type of slow-consolidating memory that remains sensitive to the amnestic cold shock two hours after conditioning, unlike the amnestic resistant memory present two hours after a single round of conditioning. We argue therefore that the two memories are distinct, and that five-round MC ostensibly does not yield ARM alone, but a labile memory as well. We suggest the term protein synthesis independent memory (PSIM) for the labile memory elicited by MC protocols, and ARM for the cold shock resistant 3-h memory after a single round of training to distinguish them. The collective evidence presented herein strongly suggests that one training round elicited ARM is not equivalent to memory yielded by five rounds and most likely ten rounds of MC, which yields ARM and PSIM; therefore, the terms should not be used interchangeably.

Even though a number of genes and molecular pathways have been reported to function in ARM, evidence supporting their involvement comes largely from one of the two assays, either 3-h memory after a cold shock (ARM) or after MC, with few tested in both assays [4,7,13] uncovering similar defects. However, for these and others that have been characterized solely via MC protocols, the effect of cold shock 2 h post-training on 24-h memory has not been assessed, so it remains an open question as to whether defects in these mutants result from the ARM or the PSIM component. Archetypical mutants such as *radish* with clear deficits in ARM [12,14] have not been subjected to our knowledge to amnestic treatments after MC, so their reported 24-h memory deficit after 10-round MC [3], may also harbor a PSIM deficit. Furthermore, it would be interesting to investigate whether PSIM is affected or parallels the labile memory compromised in mutants like *amnesiac* [15,16]. Common molecular components of PSIM and other labile memory types would suggest at least partially overlapping molecular mechanisms, yet perhaps distinct enough to differentiate the two processes, a hypothesis currently under investigation.

## 4. Materials and Methods

### 4.1. Drosophila Culture and Strains

Cantonized *w^1118^* and *Canton S* wild-type strains were cultured in wheat flour–sugar food as previously described [4] and raised in a 12 h night/dark cycle, at 25 °C and 50% humidity.

### 4.2. Behavioral Experiments

The 2–4-day old flies were used in all experiments, which were performed at 25 °C and 55–65% humidity under dim red light. Aversive olfactory conditioning utilized 90 Volt electric foot-shocks as unconditioned stimuli (US), paired with one of the aversive odorants 5% benzaldehyde (BNZ), or 50% octanol (OCT) diluted in isopropyl myristate as conditioned stimuli (CS). One training cycle consisted of 12 CS/US pairings of 1.25 s with a 4-s interstimulus interval, followed by 30 s of rest before presenting another odor in the absence of shock. Either odor was paired with shock, while the other served as control. Massed conditioning (MC) involved five consecutive training cycles with 30 s between cycles. Spaced training was identical except the interval between cycles was 15 min. Cold-shock treatment was administered as described previously [4] at 1 min, 2, or 23 h after the final training round, as indicated in each experiment. Memory testing involved simultaneous presentation of both odors for 90 s as described [4]. To calculate Δ, the difference between labile and consolidated memories, two groups of animals were simultaneously trained with 1 round of training, and half were subjected to cold shock while the others were not. Δ was calculated as the performance difference between simultaneously trained untreated and cold-shocked animals. A similar method was used to calculate Δ for animals trained with 5 rounds MC.

### 4.3. Data Analysis

Raw data analysis was performed with the JMP7 software (SAS Institute Inc., Cary, NC, USA). Statistical comparisons were performed as detailed in the figure legend. Comparison between two groups was carried out with ANOVA and subsequent LSM-planned comparisons or, in cases of different variances between groups, unpaired parametric Welch’s *t*-test when variances of the measurements were unequal. Statistical details are presented in Table 1. Graphs were created with the GraphPad Prism 8.0.1 software and show means ± SEM.

## Author Contributions

Conceptualization, A.B. and E.M.C.S.; methodology: A.B. and E.M.C.S.; formal analysis: A.B.; writing—original draft preparation, A.B.; writing—review and editing: E.M.C.S.; supervision: E.M.C.S. All authors have read and agreed to the published version of the manuscript.

## Figures and Tables

**Figure 1 ijms-23-06407-f001:**
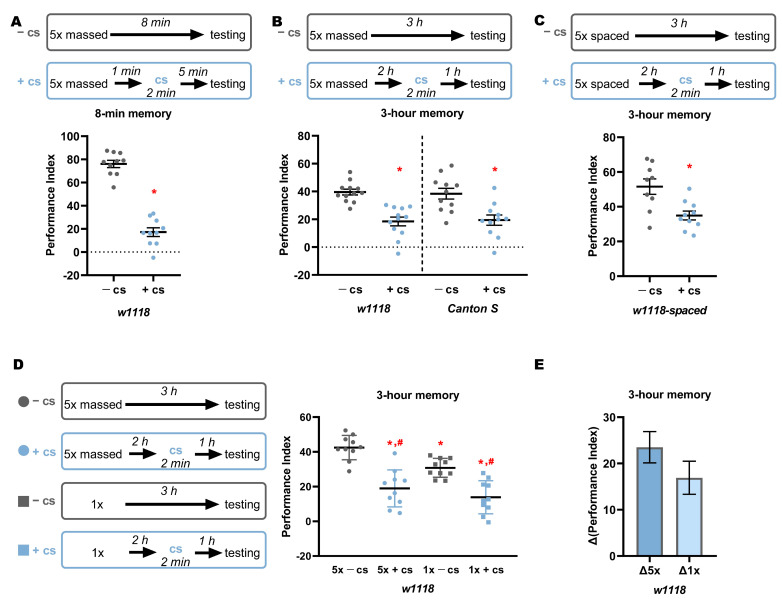
Memory acquired by massed training is cold-shock sensitive. The graphs show mean performance ± SEM consequent to the treatments detailed above. Star and pound symbols indicate significant differences as detailed below. (**A**) An 8-min (immediate) memory of *w^1118^* animals is inhibited by a 2-min by cold shock (ANOVA F_(1,20)_ = 144.82, *p* < 4.8 × 10^−10^). (**B**) Three-hour memory produced by massed training is significantly reduced by cold shock treatment in both *w^1118^* (ANOVA F_(1,24)_ = 30.74, *p* < 1.4 × 10^−5^) and *Canton S* (ANOVA F_(1,22)_ = 12.41, *p* = 0.002) animals. (**C**) Three-hour memory of *w^1118^* animals after spaced training is significantly affected by cold shock (ANOVA F_(1,19)_ = 10.87, *p* = 0.004). (**D**) Three-hour memory in *w^1118^* flies after one training round is different from memory formed after five consecutive rounds [(ANOVA F_(3,40)_ = 23.05, *p* = 1.7 × 10^−8^). Subsequent analysis using LSM-planned comparisons revealed that the difference between the two groups is indeed significant (*p* = 0.003, star)]. The performance of cold shocked flies was significantly different from that of untreated animals after one or five MC rounds (*p* < 0.0001, pound). However, memories resilient to cold shock were not significantly different (*p* = 0.1856). (**E**) Differences in indexes shown in (**D**) between treated and non-treated groups that were simultaneously trained with five or one training round are not significantly different (ANOVA F_(1,20)_ = 1.79, *p* = 0.1965).

**Figure 2 ijms-23-06407-f002:**
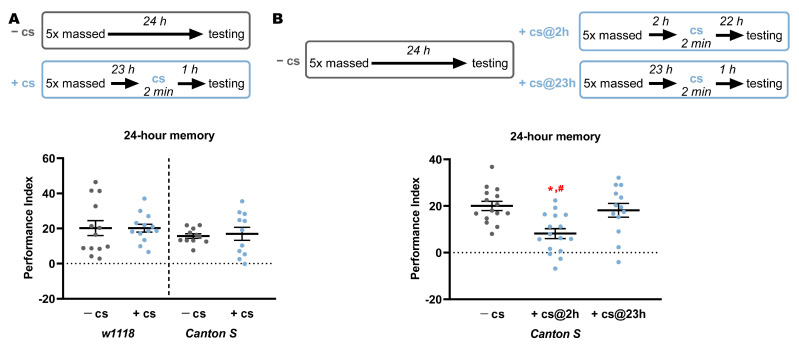
Memory acquired by massed training is resistant to cold shock at 23 h. Graphs show mean performance ± SEM consequent to the treatments detailed above. Star and pound symbols indicate significant differences as detailed below. (**A**) The 24-h memory elicited by MC is not affected by cold shock delivered 23 h post-training. Because variances were different, the unpaired parametric Welch’s *t*-test was used to compare means. (*t* = 0.0039, df = 18.23, *n* = 13. *p* = 0.9969 for *w^1118^* and *t* = 0.3129, df = 12.50, *n* = 11. *p* = 0.7595 for *Canton S*). (**B**) The 24-h memory elicited by MC is compromised if animals are cold shocked 2 h post-training, but not if cold shock is delivered 1 h before testing. [ANOVA F_(2,44)_ = 7.992, *p* = 0.001. Subsequent comparisons using LSM-planned comparisons to non-cold shocked animals revealed significant differences in the performance of animals cold-shocked 2 h post-training (*p* = 0.0006, star) and from those cold-shocked at 23 h (*p* = 0044, pound), but not from animals cold shocked 23 h post-training (*p* = 0.5759)].

**Table 1 ijms-23-06407-t001:** Statistical comparisons. Note the final two comparisons in analysis regarding Figure 1D and Figure 2B are not with the relevant control, but rather between treatments. Bold numbers indicated significant *p* values.

Group	Mean ± SEM	*p* Value
**Figure 1A**	ANOVA F_(1,20)_ = 144.8201 ***p* = 4.8 × 10^−10^**	
w1118 (−cs)	76.04 ± 3.16	
w1118 (+cs)	17.22 ± 3.73	**<0.0001**
**Figure 1B**	ANOVA F_(1,24)_ = 30.7419 ***p* = 1.4 × 10^−^**^5^	
w1118 (−cs)	39.52 ± 2.08	
w1118 (+cs)	18.46 ± 3.18	**<0.0001**
	ANOVA F_(1,22)_ = 12.4188 ***p* = 0.0021**	
Canton S (−cs)	38.39 ± 3.89	
Canton S (+cs)	19.42 ± 3.72	**0.0021**
**Figure 1C**	ANOVA F_(1,19)_ = 10.8714 ***p* = 0.0043**	
w1118 (−cs)	51.48 ± 4.49	
w1118 (+cs)	34.93 ± 2.59	**0.0043**
**Figure 1D**	ANOVA F_(3,40)_ = 23.0502 ***p* = 1.7 × 10^−8^**	
w1118 (5x − Ics)	42.46 ± 2.22	
w1118 (5x + cs)	18.94 ± 3.37	**<0.0001**
w1118 (1x − cs)	30.78 ± 1.72	**0.0037**
w1118 (1x + cs)	13.86 ± 3.01	**<0.0001**
w1118 (1x − cs)	30.78 ± 1.72	
w1118 (1x + cs)	13.86 ± 3.01	**<0.0001**
w1118 (5x + cs)	18.94 ± 3.37	
w1118 (1x + cs)	13.86 ± 3.01	**0.1856**
**Figure 1E**	ANOVA F_(1,20)_ = 1.7990 *p* = 0.1965	
w1118 (Δ5x) = (5x − cs) − (5x + cs)	23.52 ± 3.38	
w1118 (Δ1x) = (1x − cs) − (1x + cs)	16.92 ± 3.58	0.1965
**Figure 2A**	ANOVA F_(1,22)_ = 1.57 × 10^−5^ *p* = 0.9969	
	Welch-corrected *t* = 0.00395711, df = 18.23	
w1118 (−cs)	20.22 ± 4.26	
w1118 (+cs)	20.20 ± 2.26	0.9969
	ANOVA F_(1,26)_ = 0.0979 *p* = 0.7576	
	Welch-corrected *t* = 0.3129, df = 12.50	
Canton S (−cs)	15.71 ± 1.33	
Canton S (+cs)	16.94 ± 3.72	0.7595
**Figure 2B**	ANOVA F_(2,44)_ = 7.9924 ***p* = 0.0012**	
Canton S (−cs)	20.02 ± 1.94	
Canton S (+cs@2 h)	8.15 ± 2.11	**0.0006**
Canton S (+cs@23 h)	18.13 ± 2.95	0.5759
Canton S (+cs@23 h)	18.13 ± 2.95	
Canton S (+cs@2)	8.15 ± 2.11	**0.0044**

## Data Availability

All relevant data are presented within this manuscript.

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
