# Peer review of "Cold Shock Disrupts Massed Training-Elicited Memory in Drosophila"

_ijms, 2022, doi:10.3390/ijms23126407_

Round 1
Reviewer 1 Report
The identification of different memory processes in Drosophila has come from environmental disruptions such as cold shock and protein synthesis inhibitors, or from the analysis of mutants and transgenes that disrupt or eliminate specific signaling pathways or cellular processes. In this manuscript, the authors use Cold-shock with different training regimes to more clearly delineate the extent and nature of Anesthesia Resistant Memory. Specifically, their data suggest an additional form of protein synthesis independent memory (PSIM) that is elicited after 5 training trials.
Overall, the experiments in Figure 1 are a nice extension of the parameters under which consolidated memories form after massed training and provide additional insights in memory dynamics. These experiments appear to have been performed rigorously with appropriate statistical analysis.
I have just a few minor comments:
Given that ARM was also identified as defective in the radish mutant, and that amnesiac mutants define an Anesthesia-sensitive memory component, which may parallel the non-ARM memory the authors found after 5 training trials, I believe it would be very helpful for the authors to briefly discuss and speculate on the potential roles of these mutants (or others, such as drk or 5HT1a) in PSIM.
The conversion of ASM to ARM after a single training trial is time-dependent. In the experiments presented in this manuscript, the parameter of time is altered by the addition of four additional training trails. I am not concerned by this slight shift in timing, but it should be explicitly acknowledged by the authors. The authors should also more clearly state in the method that the cold shock was delivered two hours after the final training trial.
In Methods please state that the Welch’s t-tests were used when the variances of the measures were unequal, assuming that was the criteria.
Line 68: delete “is”
Lines 24 to 36 appear to have an alternative spacing compared to the rest of the document.
Table 1:
ANOVAs above data are not aligned, especially for Figures 1 B and 1D.
The Line separating analyses in Fig 1E is not straight
Author contributions:
Line 139 and 140 are directions for this selection and do not belong in the manuscript.
Funding:
Line 141, remove “Please add:”
The authors acknowledge support from Fondation Santé, but no external funding was received. Is this correct?
Author Response
Given that ARM was also identified as defective in the radish mutant, and that amnesiac mutants define an Anesthesia-sensitive memory component, which may parallel the non-ARM memory the authors found after 5 training trials, I believe it would be very helpful for the authors to briefly discuss and speculate on the potential roles of these mutants (or others, such as drk or 5HT1a) in PSIM.
>>>we have added relevant comments to the discussion easily identified in the marked up version of the manuscript
The conversion of ASM to ARM after a single training trial is time-dependent. In the experiments presented in this manuscript, the parameter of time is altered by the addition of four additional training trails. I am not concerned by this slight shift in timing, but it should be explicitly acknowledged by the authors. The authors should also more clearly state in the method that the cold shock was delivered two hours after the final training trial.
>>> we have made explicit statements regarding cold shock delivery in our 5MC protocol relative to the classic one after one round of training.
In Methods please state that the Welch’s t-tests were used when the variances of the measures were unequal, assuming that was the criteria.
>>> indeed these were the criteria and we have stated so in the statistics section of Materials and Methods
Reviewer 2 Report
Paper by Bourouliti at al. provides a novel fact allowing to better differentiate between two types of memory in Drosophila: memory after a single olfactory conditioning episode and a massed conditioning-elicited memory (after olfactory conditioning episodes repeated fast 5 times in a row). It shows that massed conditioning-elicited memory can be partially disrupted by a cold shock 2 hours after training, although memory after a single olfactory conditioning episode cannot be disrupted by a cold shock (therefore this type of memory was named ARM, Anesthesia Resistant Memory).
These two memory-inducing procedures were established and studied for over 30 years and were believed to induce memory consolidation by the same mechanism. Bourouliti at al. disproved this belief by showing that memory consolidation after massed conditioning is not the same as in ARM. Authors propose new term Protein Synthesis Independent Memory (PSIM) for memory elicited by massed conditioning protocols. Thus, the paper provides an important and novel result for the field of neuroscience.
Paper is mostly written clearly. Methods are adequate and results of experiments substantiate conclusions. Nevertheless, paper requires improvements.
1) First of all, the paper presents only “5-round MC” results (Fig.1). However, the main conclusions of the paper are based on the result about ARM (memory after a single olfactory conditioning episode) from previous studies, as authors compare the effect of cold shock on these 2 types of memory. These results are not shown in the current paper text, which makes it impossible to directly compare Performance Index in ARM with that in “5-round MC”. Consequently, it’s impossible without looking into cited papers to see to what % ARM is the part of “5-round MC” memory.
Ideally the authors should present their own results of PI after a single olfactory conditioning episode (ARM -cs, +cs) to directly compare.
Another way to solve this issue is to add a graph summarizing ARM results from previous papers together with the mean PIs obtained in the current study for “5-round MC” and results on “5-round MC” from previous studies to show that control PI in this study is consistent with the previous studies. Such graph will allow to see that “5-round MC” leads to better memory (higher PI) compared to single conditioning and if cold shock disruption puts it to “ARM base” or not.
2) The main conclusion is formulated slightly differently in different parts of the paper, which leads to confusion. In results section it reads:
(Line 54) “Therefore, memory elicited by 5 rounds of MC does not yield solely ARM, but rather both cold shock-sensitive and cold shock-insensitive memory components.”
This formulation reflects the experimental results in a most correct way.
However, the phrase in the discussion section doesn’t say that ARM is a part of 5-round MC, which confuses the reader:
(Line 103-105): “We argue therefore that the two memories are distinct and that 5-round MC ostensibly does not yield ARM.”
Lines 103-105 should be rewritten in a way that does not obscure the actual result shown in the paper, namely, that “5 rounds MC” switches on some additional mechanism of memory consolidation, which is added to ARM. This conclusion seems logical, because “5 rounds MC” contains also the first single conditioning episode, which should lead to ARM. How otherwise memory established after the first conditioning episode can be disrupted by the following 4 rounds of MC?
The rest of the text should be also checked to be sure to avoid obscure formulations.
3) Introduction should be improved:
-Add explanation of why “5 round MC” is investigated. Is it the most established protocol used in previous studies?
- Line 44: The way of massed conditioning “5 MC rounds” has to be explained in the first mentioning of this term.
- Last paragraph of introduction has to contain the result of the study in short as a last sentence.
-References should be checked and used more carefully. For example, Line 30 “ARM appears to consolidate relatively rapidly as it is partially labile minutes after conditioning [5] and stable by 2 hours post-training [6].” cites only refs 5 and 6, although the same result was shown in the ref 3, which is a much older paper than 5 and 6. Why is that? Results from references should be provided in more detail if the use of 5 and 6 instead of 3 is justified.
4) Minor comments:
Line 58: “LTM “– long term memory? - explain abbreviation.
Line 58: “Spaced conditioning” – the term appears first time in the text without any explanation. Definition should be added in the introduction or here.
Author Response
First of all, the paper presents only “5-round MC” results (Fig.1). However, the main conclusions of the paper are based on the result about ARM (memory after a single olfactory conditioning episode) from previous studies, as authors compare the effect of cold shock on these 2 types of memory. These results are not shown in the current paper text, which makes it impossible to directly compare Performance Index in ARM with that in “5-round MC”. Consequently, it’s impossible without looking into cited papers to see to what % ARM is the part of “5-round MC” memory.
Ideally the authors should present their own results of PI after a single olfactory conditioning episode (ARM -cs, +cs) to directly compare.
>>> since we had the flies ready and we were given a bit of extra time to resubmit, we opted to perform the experiment of direct comparison of labile and consolidated memories elicited by one versus 5 rounds of conditioning. The results are presented in revised Figure 1D and E. We present a significant difference in total memory elicited by 5MC versus 1 round, but not a significant difference in consolidated memory/ARM.
Because of the addition on these two panels we found it prudent to break up the Figure of the original manuscript into Figure 1 and Figure 2. We hope this is OK.
2) The main conclusion is formulated slightly differently in different parts of the paper, which leads to confusion. In results section it reads:
(Line 54) “Therefore, memory elicited by 5 rounds of MC does not yield solely ARM, but rather both cold shock-sensitive and cold shock-insensitive memory components.”
This formulation reflects the experimental results in a most correct way.
However, the phrase in the discussion section doesn’t say that ARM is a part of 5-round MC, which confuses the reader:
(Line 103-105): “We argue therefore that the two memories are distinct and that 5-round MC ostensibly does not yield ARM.”
>>we have addressed these concerns and we thank the reviewer for pointing out the inconsistency. These changes are obvious in the marked up version of the manuscript.
In addition we have expanded the introduction to address the suggestions of the reviewer and have detailed our protocol and why we use that particular method.
we have also expanded and augmented the references and included the change suggested (correctly and we are thankful for pointing it out) in failing to site a relevant paper while referencing others.